# Artificial Intelligence-Assisted Renal Pathology: Advances and Prospects

**DOI:** 10.3390/jcm11164918

**Published:** 2022-08-22

**Authors:** Yiqin Wang, Qiong Wen, Luhua Jin, Wei Chen

**Affiliations:** 1Department of Nephrology, The First Affiliated Hospital, Sun Yat-sen University, Guangzhou 510080, China; 2NHC Key Laboratory of Clinical Nephrology (Sun Yat-sen University) and Guangdong Provincial Key Laboratory of Nephrology, Guangzhou 510080, China

**Keywords:** renal pathology, digital imaging, image interpretation, machine learning, artificial intelligence, kidney diseases

## Abstract

Digital imaging and advanced microscopy play a pivotal role in the diagnosis of kidney diseases. In recent years, great achievements have been made in digital imaging, providing novel approaches for precise quantitative assessments of nephropathology and relieving burdens of renal pathologists. Developing novel methods of artificial intelligence (AI)-assisted technology through multidisciplinary interaction among computer engineers, renal specialists, and nephropathologists could prove beneficial for renal pathology diagnoses. An increasing number of publications has demonstrated the rapid growth of AI-based technology in nephrology. In this review, we offer an overview of AI-assisted renal pathology, including AI concepts and the workflow of processing digital image data, focusing on the impressive advances of AI application in disease-specific backgrounds. In particular, this review describes the applied computer vision algorithms for the segmentation of kidney structures, diagnosis of specific pathological changes, and prognosis prediction based on images. Lastly, we discuss challenges and prospects to provide an objective view of this topic.

## 1. Background

Artificial intelligence (AI) and machine learning (ML) have triggered a vigorous technological revolution in the medical field. The use of AI algorithms provides cutting-edge guidance for clinical practice, including medical image analysis, smart diagnosis, curative effect evaluation, and prognosis prediction [1,2]. AI technology is also regarded as a useful tool to improve the diagnostic efficiency and accuracy of renal pathologies. Both the latest progress and existing problems in the field of AI-based renal pathology are summarized in this review, providing references and new insights for subsequent research.

## 2. Current Concepts of AI

AI refers to imitating natural intelligence perception and decision-making strategies in computers to take optimal actions that are not explicitly programmed for the goals [3,4]. ML is a branch of AI concerned with algorithms necessary to train a model to acquire, integrate, and learn new knowledge based on large-scale observations and empirical data [5]. ML has been broadly applied in complicated tasks, including image analysis and automatic natural speech processing [6,7,8]. Deep learning is a subclass of ML characterized by an algorithm that can combine raw inputs into multi-layered neural networks with millions of neuron-like units, which mimics the human brain’s ability to conduct data interpretation and significantly improve overall performance [9,10]. Artificial neural networks (ANNs), which involve multiple layers between input and output, are known as deep neural networks (DNNs). The most commonly used DNN types are multilayer perceptron (MLP), convolution neural networks (CNNs), and recurrent neural networks (RNNs) (details are shown in Figure 1).

## 3. AI Image Processing Workflow

AI image processing technology with advanced algorithms obtained by ML and computer vision can process large volumes of pictures easily and quickly. An AI image processing workflow can be divided into three phases: (1) data selection, collection, and annotation; (2) image pre-processing and model development; and (3) model verification and data fusion (Figure 2).

### 3.1. Data Selection, Collection, and Annotation

In past decades, pathological diagnoses have been made based on small histological images acquired using light microscopy. With the development of whole-slide imaging (WSI) technology, it is possible to assemble the captured sequential images into a high-resolution virtual slide, providing convenient access for AI image processing and avoiding deterioration of staining quality over time [14]. Since this is the first step for AI training, the quality of data selection and collection is critical for the final performance of image processing. Therefore, unqualified images with blurry vision, poor staining, and air bubbles should be checked and discarded to avoid misleading AI image processing [15].

According to the dependency on the label, AI training algorithms can be established using supervised, unsupervised, or semi-supervised learning approaches. Supervised learning creates an accurate prediction model mainly based on a well-annotated dataset. An associated label is required for each input data point in the process of supervised learning, whereas unsupervised learning receives unlabeled data and identifies patterns automatically [16]. Semi-supervised learning is developed by mixing the abilities of supervised and unsupervised learning. A partially trained model is initially constructed using a small subset of labeled data, which can be used to “label” the remaining unlabeled parts [17].

### 3.2. Image Pre-Processing and Model Development

Image pre-processing usually includes image normalization and augmentation, which is necessary before patch delivery into the model. Feature normalization, which adjusts all features to the same scale, ensures that the features with various ranges contribute approximately proportionately to the final response. Image augmentation is a technique that is used to manually enlarge the size of the dataset to avoid overfitting when very few data samples are used for deep learning [18].

After image data preparation, an ML model is constructed and then used for feature extraction and data analysis. There are two main approaches for feature extraction: handcrafted, or unsupervised extraction [19]. Handcrafted features are characterized by low-dimensional, intuitive variables including color, shape, statistical, and textural features that allow for explicit modeling of morphology [20]. Unsupervised feature extraction is used for redundancy minimization of the large amount of training data with a deep learning method [19,21].

There are three basic tasks of image analysis: image classification, object detection, and structure segmentation [22]. Image classification refers to the assignment to attach a label to an image, for which a corresponding class is selected. Object detection can recognize and localize the objects using bounding boxes. Unlike the two tasks mentioned above, structure segmentation provides details of a given object at a pixel level. Images are simplified as the congregation of different pixel groups via ML with assigned labels, and thereby the exact outline of the object can be drawn accordingly [23].

### 3.3. Model Verification and Data Fusion

To evaluate the performance of a model, a validation dataset is required, consisting of images unknown to the machine [24]. The common metrics, such as accuracy, specificity, sensitivity, F1-Measure, a receiver operating characteristic (ROC) curve, and an area under the ROC curve (AUC), are used to estimate the ability and generalizability of the model [25]. To avoid occupying the memory of the local computer, it is better to further test the model using the external validation data, which are different from the data in the training dataset or the validation dataset. 

To improve the prediction performance of the model and provide a more personalized diagnosis and prognosis, it is suggested that comprehensive information be incorporated into the model, such as clinical history, pathological examination, genomics, and radiomic data of patients [26,27]. For instance, Mobadersany et al. combined glioma histological images with genomic biomarkers to form a unified prognostic prediction model, the accuracy of which was even superior to that of the current paradigm [28].

## 4. Application of AI in Nephropathology

In recent years, chronic kidney disease (CKD) has become a major worldwide health issue with increasing incidence and prevalence. According to statistical reports, the global prevalence of CKD in 2019 was approximately 13.4%, and the number of patients with end-stage renal disease (ESRD) was estimated to be in the range from 4.902 to 7.803 million [29]. Therefore, early identification of disease etiology is recognized as a top priority for nephrologists to carry out targeted treatment and delay progression to CKD. To this end, as part of routine clinical practice, renal biopsy is an indispensable procedure that provides an objective basis for the determination of a definite diagnosis, prognosis, and appropriate treatment plan [30]. However, the current diagnosis made by renal pathology mainly depends on the assessment of a renal pathologist, which is not only time-consuming and labor-intensive but also involves subjectivity and relatively poor reproducibility [31]. Although pathologists receive the same systematic training and use the same standardized guidelines, diagnostic discrepancies still exist due to different visual perceptions, data processing habits, and judgment preference [32]. AI-based state-of-the-art technology might provide a possible solution for this problem. The various applications of AI algorithms in renal pathology are summarized in Table 1, Table 2 and Table 3.

### 4.1. Detection and Segmentation of Kidney Structures

AI applications in the field of renal pathology started with imaging detection and segmentation of glomeruli because of their distinguishing features compared to other renal structures. Glomerular damage accounts for a substantial proportion of progressive CKD, leading to a decline in kidney function over time and even ESRD [33]. The number of normal glomeruli and the incidence of glomerulosclerosis are routinely assessed by pathological examination of kidney biopsies [34], indicating the risk of progression of glomerular diseases, such as glomerulonephritis and IgA nephropathy. However, a recent study (2016) showed that the number of glomeruli and the ratio of glomerulosclerosis measured by traditional light microscopy are inaccurate compared to those in WSI analysis. The errors were positively correlated with the sum of glomeruli. Therefore, the introduction of AI-assisted WSI analysis might be helpful to promote more precise evaluation of glomeruli [35].

In 2018, Simon et al. developed a support vector machine (SVM) model that could automatically identify normal glomeruli in mouse tissue samples. According to the image features provided by the local binary pattern (LBP), an effective texture descriptor for images, the model achieved a high accuracy of 90% and a recall rate of 70%. Moreover, it can be used for glomeruli detection in rat and human tissues regardless of staining conditions [36].

Apart from the identification of normal glomeruli, an AI application has also been reported in detecting glomerular lesions. Glomerular proliferative lesions, which are characterized by the increased number of cells in the glomeruli, mesangial area, or the capillary lumen, are considered the activity indicators for IgA nephropathy and lupus nephritis [37,38]. To identify abnormalities in glomerular proliferation, Chagas et al. proposed a new CNN network in a combination of SVM for the assessment of three sub-classifications of hypercellularity (mesangial proliferation, endocapillary hyperplasia, and mixed types) with an accuracy of 82% [39]. Moreover, glomerulosclerosis is characterized by sclerosis of various extents ranging from the segment to the entire glomerulus [40], indicating the extent of chronic kidney damage with a weak response to therapy of immunosuppression [41]. In kidney transplantation, the percentage of global glomerulosclerosis is also considered a determining factor in graft acceptance. Thus, there is a need to evaluate the status of global glomerulosclerosis carefully before implantation [42]. For example, a robust CNN network was established to segment and classify the various glomerular pathologic changes in the tissue slides from renal biopsies with diverse staining backgrounds. The image datasets in that study included 1123 snapshots and 348 WSIs. This network was trained to classify glomeruli into three categories, including normal glomeruli, sclerotic glomeruli, and glomeruli with other lesions. Using this network, the F1-score of the subgroups achieved 0.68–0.90 in the snapshot group, and the score reached a comparable level (0.75–0.83) in the WSI group [43]. In addition, new AI-assisted technique applications have been reported in recent studies, such as non-label classification and fine-grained characterization of glomerulosclerosis in renal biopsy pathological images [44,45].

Aside from global sclerosis and proliferative changes, AI-assisted identification of other lesions, such as crescents, also raised the interest of nephropathologists. To break the limitations of monotype change and to explore more types of pathological features, two CNN-based models were constructed, which can classify 12 and 9 pathological features of glomeruli, respectively. The performance of both models achieved moderate-to-high levels [46,47]. However, these two models are only suitable for analyzing the images of PAS-stained biopsies and may neglect certain characteristics shown by other staining methods. Therefore, to reach a more accurate and specific diagnosis, those images with different staining need to be merged for model training. Moreover, other types of images, such as immunofluorescence (IF) snapshots, have been suggested for integration. It has been reported that the appearance of immunoglobin deposition located in the area of the glomerular lesion can be automatically classified by deep learning approaches with high accuracy of more than 95% [48]. Thus, the combination of IF and light microscopic data might be realized soon.

In addition to the importance of glomeruli as an indispensable part of the kidney, pathological changes, such as developmental abnormalities, inflammation, and fibrosis, in other subtle structures including tubules and arteries can also be detected in digital images [49]. For example, the scores for interstitial inflammation, tubulitis, and intimal arteritis are included in the Banff classification reporting system, which is an international consensus system for renal allograft pathology evaluation [50]. A new trend in the field is to develop AI models to evaluate whole renal structures, not limited to glomeruli. One report focused directly on the segmentation of human kidney structures using 40 cases of PAS-stained WSIs made from kidney transplant biopsies for model training. The results revealed a high average Dice coefficient weighted by all classes of structure, regardless of the centers (0.80 and 0.84 on 2 datasets) and the source of samples (biopsy/nephrectomy) [31]. Further analysis showed that the AI’s ability to identify glomeruli, tubules, and interstitium of the kidney was top-ranked, while its ability to recognize atrophic tubules and empty Bowman’s capsule was less satisfying. Significant correlations were also found between quantifications of CNN segmentation and visually scored Banff classification, indicating the applicability of CNN in automatic routine evaluation for transplant kidney conditions.

Moreover, as a special “structure” in kidney, cancer masses can also be automatically classified and evaluated by grades with the help of deep learning algorithms. Fenstermaker et al. developed a CNN model based on H&E-stained WSIs from 42 renal cell carcinoma (RCC) specimens to distinguish normal tissue and 3 histology cancer subtypes including clear cell, chromophobe, and papillary carcinoma [51]. The accuracy of the model could reach as high as 99% in tumor tissue identification and 97.5% in RCC subtype classification. In addition, the model also predicted prognosis-associated Fuhrman grade according to nuclear size and polymorphism with a high accuracy of 98.4%. Thus, the results indicate that by highlighting the region of interest in advance and presenting their judgements for reference, artificial intelligence methods are expected to improve the accuracy and efficiency of kidney cancer diagnosis in the future. The AI applications in the identification of different renal structures are illustrated in detail in Table 1.

**Table 1 jcm-11-04918-t001:** AI-aided identification of renal structure.

Object	Author	Year	Task	Methods	Slides	Main Results	Ref.
**Normal glomeruli**	Simon et al.	2018	Localization of glomeruli	CNN, SVM	15 WSIs, healthy mice (H&E) 15 WSIs, STZ-mice (H&E) 15 WSIs, rat (CR, H&E, Jones, PAS, and Gömorri trichrome) 25 WSIs, DN patients (PAS)	Glomerular detection in mouse: precision: >90%; recall: >70%	[36]
	Bukowy et al.	2018	Localization of glomeruli with trichrome-staining	Alexnet CNN	87 WSIs, rat (Gömöri or Masson trichrome)	Average precision: 96.94%; recall: 96.79%	[52]
	Sheehan et al.	2018	Segmentation and quantification of glomeruli	Ilastik	738 images, mice (PAS)	Precision: 98.4%; recall: 95.2%, F-score: 96.0%	[53]
	Wilbur et al.	2021	Detection of glomeruli of four different stains across institutions	CNN	284 WSIs, human (H&E, PAS, PASM, trichrome)	Sensitivity: intra-institutional: 90–93%; interinstitutional: 77%; combined: 86% Modified specificity: intra-institutional: 86–98%; interinstitutional: 97%; combined: 92%	[54,55]
**Proliferative glomeruli**	Chagas et al.	2020	Binary or multiple classification of hypercellularity	CNN, SVM	811 images, human (H&E, PAS)	Binary classification: average accuracy: nearly 100% Multiple classification: average accuracy: 82%	[39]
	Barros et al.	2017	Segmentation and classification of glomeruli w/ or w/o proliferative changes	kNN	811 images, human (H&E, PAS)	Generalization set: precision: 92.3%; recall: 88.0%; accuracy: 88%	[56]
**Sclerotic glomeruli**	Kannan et al.	2019	Classification of normal and sclerosed glomeruli	Inception v3 CNN	171 WSIs, human (trichrome)	Accuracy: 92.67% ± 2.02%; kappa: 0.8681 ± 0.0392	[34]
	Jiang et al.	2021	Detection, classification, and segmentation of glomeruli into three categories	Cascade mask region-based CNN	1123 snapshots, human (H&E, PAS, PASM, Masson) 348 WSIs, human (H&E, PAS, PASM, Masson)	Snapshot group: F1-score: total glomeruli, GN, global sclerosis, and glomerular with other lesions (0.914, 0.896, 0.681, 0.756) WSI group: F1-score: total glomeruli, GN, global sclerosis, and glomerular with other lesions (0.940, 0.839, 0.806, 0.753)	[43]
	Lutnick et al.	2020	Label-free classification of glomeruli by Tervaert class and the presence of sclerosis	VAE-GAN	1193 individual glomeruli (H&E, PAS) 121 WSIs, human (PAS)	Cohen’s kappa values: Tervaert class: 0.87 sclerosis: 0.78	[44]
	Lu et al.	2022	Quantification and subtype classification of global glomerulosclerosis	Transfer learning	7841 globally sclerotic glomeruli of three distinct categories	Pretrained dataset: F1-score: 0.778 External dataset: AUC: 0.994	[45]
	Bueno et al.	2020	Semantic and classification of normal and sclerosed glomeruli	SegNet-VGG19+ AlexNet CNN	47 WSIs, human (PAS)	Accuracy: 98.16% F1-score: 0.994	[57]
	Gallego et al.	2021	Classification of normal and sclerosed glomeruli	U-Net CNN	51 WSIs, human (PAS, H&E)	F1-score PAS: normal glomeruli: 97.5%; sclerosed glomeruli: 68.8% H&E: normal glomeruli: 90.8%; sclerosed glomeruli: 78.1% Average: normal glomeruli: 94.5%; sclerosed glomeruli: 76.8%	[58]
	Francesco et al.	2022	Classification of sclerotic and non-sclerotic glomeruli	IBM Watson	26 WSIs, human (PAS)	Mean accuracy: 99%	[59,60,61]
	Marsh et al.	2018	Classification of non-sclerosed and sclerosed glomeruli	VGG16 CNN	48 WSIs, human (frozen sections: H&E)	Non-sclerosed glomeruli: precision: 81.3%; recall: 88.5%; F1-Score: 84.8% Sclerosed glomeruli: precision: 60.7%; recall: 69.8%; F1-score: 64.9%	[62]
	Li et al.	2021	Quantification of non-sclerotic and sclerotic glomeruli	U-Net CNN	258 WSIs, human (frozen sections)	Non-sclerosed glomeruli: Dice similarity coefficient: 0.90; recall: 0.90; F1-score: 0.93; precision: 0.96 Sclerosed glomeruli: Dice similarity coefficient: 0.93; recall: 0.87; F1-score: 0.96; precision: 0.81	[63]
	Marsh et al.	2021	Quantification of percent global glomerulosclerosis	VGG16 CNN	149 WSIs, human (frozen and permanent sections: H&E)	Higher correlation with annotations (r = 0.916; 95% CI, 0.886–0.939) than on-call pathologists (r = 0.884; 95% CI, 0.825–0.923) Lower model prediction error for single levels (RMSE, 5.631; 95% CI, 4.735–6.517) than on-call pathologists (RMSE, 6.523; 95% CI, 5.191–7.783) Decreased the likelihood of unnecessary organ discard by 37% compared with pathologists	[64]
**Glomeruli with multiple pathological changes**	Weis et al.	2022	Classification of 9 glomerular structural changes	CNN	23,395 glomerular images, human (PAS)	Kappa-values: 0.838–0.938	[46]
	Yamaguchi et al.	2021	Classification of glomerular images of 12 features	ResNet50 CNN	293 WSIs, human (PAS)	ROC–AUC: 0.65–0.98. (“capillary collapse”: 0.98)	[47]
	Zhang et al.	2022	Segmentation of glomeruli and classification of the deposition pattern in immunofluorescence image	U-Net, MANet	4779 images, human (IF)	Deposition region: accuracy: 98% Deposition appearance: accuracy: 95% Label fusion: accuracy: >90%	[48]
	Uchino et al.	2020	Classification of glomeruli of 7 pathological changes	InceptionV3 CNN	283 WSIs, human (PAS, PASM)	Global sclerosis: AUC: PAS: 0.986; PASM: 0.983 Other pathological findings: AUC: 0.59–0.87 (close to those of nephrologists)	[65]
	Yang et al.	2021	Detection, classification, lesion identification of glomerular disease	Mask R-CNN, LSTM RNN, ResNeXt-101	Detection: 1379 slides, human (H&E, PAS, TRI, PAM) Classification: 653 cases, human	Detection: F1-scores: up to 0.944 Classification: accuracies: up to 0.940 Lesion identification: AUC: up to 0.947	[66]
	Nan et al.	2022	Classification of five subcategories of IgAN glomerular lesions	UAAN	400 WSIs, human (PAS)	Accuracy: 93.0% Fl-score: 92.9%	[67]
**Other kidney structures**	Hermsen et al.	2019	Multiclass segmentation of kidney biopsies	U-Net CNN	132 WSIs, human (PAS)	Weighted mean Dice coefficients of all classes: 0.80–0.84 Mean intraclass correlation coefficient (pathologists versus the network): 0.94	[31]
	Sheehan et al.	2019	Identification of histological differences between mice of different genotypes according to segmentation of kidney structure	AlexNet DNN, SVM	90 WSIs, mice (PAS)	Identification of previously neglected histologic features, including vacuoles, nuclear count, and proximal tubule brush border integrity, to distinguish mice of different genotypes	[68]
	Bouteldja et al.	2021	Segmentation of kidney tissue	U-Net CNN	168 WSIs, healthy and diseased mouse, pig, marmoset, bear and rat, human (PAS)	Multiclass segmentation performance was very high in all murine disease models (Dice score: 73.5–98.8) and in other species (Dice score: 76.6–99)	[69]
	Jayapandian et al.	2021	Segmentation of histologic structures in multi-stained kidney biopsies	U-Net CNN	459 WSIs, human (H&E, PAS, TRI, SIL)	F-scores: PAS (optimal): glomerular tufts: 0.93; glomerular tuft plus Bowman’s capsule: 0.94; proximal tubules: 0.91; distal tubular segments: 0.93; peritubular capillaries: 0.81; arteries and afferent arterioles: 0.85	[70]
	Govind et al.	2021	Label-free identification and quantification of podocyte	Cloud-based AI	122 WSIs, mouse, rat, and human (PAS)	Sensitivity/specificity: mouse: 0.80/0.80; rat: 0.81/0.86; human: 0.80/0.91	[71]
**Renal cell carcinoma**	Michael Fenstermaker et al.	2020	Identification and evaluation of renal cell carcinoma	CNN	12,168 RCC samples, human	Accuracy: normal parenchyma vs. RCC: 99.1%; clear cell, papillary, and chromophobe histiotypes: 97.5%; Fuhrman grade: 98.4%	[51]
	Eliana Marostica et al.	2021	Classification and prediction of clinical outcomes in subtypes of renal cell carcinoma	Deep convolutional neural networks (DCNN)	231 slides (chRCC), 1657 slides (ccRCC), 475 slides (pRCC), human	AUC: detection of malignancy: 0.964–0.985; diagnosis of RCC histologic subtypes: 0.953–0.993	[72]
	Sairam Tabibu et al.	2019	Classification and survival prediction of renal cell carcinoma	CNN	1027 images (ccRCC), 303 images (pRCC), and 254 images (chRCC), human	Classification of RCC histologic subtypes: 94.07%	[73]
	Mengdan Zhu et al.	2021	Classification of 4 subtypes of renal cell carcinoma	Deep neural network	1074 WSIs, human	AUC: 0.97–0.98	[74]

Abbreviations: CNN: convolutional neural network; SVM: support vector machine; WISs: whole-slide images; H&E: hematoxylin and eosin; STZ: streptozocin; CR: Congo red; PAS: periodic acid–Schiff; DN: diabetic nephropathy; PASM: periodic acid–silver methenamine; kNN: k-nearest neighbor; VAE–GAN: variational autoencoder–generative adversarial network; AUC: area under the curve; CI: confidence interval; RMSE: root-mean-square error; ROC: receiver operating characteristic curve; Mask R-CNN: mask region-based convolutional neural networks; LSTM: long short-term memory; MANet: multiple attentions convolutional neural network; IF: immunofluorescence; RCC: renal cell carcinoma; chRCC: chromophobe renal cell carcinoma; ccRCC: clear cell renal cell carcinoma; pRCC: papillary renal cell carcinoma.

### 4.2. Auxiliary Diagnosis of Renal Pathological Changes

#### 4.2.1. Renal Interstitial Fibrosis

Renal interstitial fibrosis is the main pathological change in the period of end-stage renal failure, which is closely associated with the progression of various CKDs and the prognosis of kidney transplantation [75]. In addition, interstitial fibrosis (IF) and tubular atrophy (TA) are also regarded as the featured histologic changes of chronic allograft injury (CAI). The severity of CAI indicates a poor prognosis for renal allograft survival [76]. Therefore, early diagnosis and intervention of renal interstitial lesions are of great significance in delaying the loss of renal function.

Currently, the pathology of kidney biopsies is still a gold standard for diagnosing renal fibrosis [77]. Kidneys with interstitial fibrosis may have fibrous changes in different structures, such as the interstitium with remarkable inflammation, glomeruli with diffused fibrosis, atrophy tubules, and thickened renal arterioles [78]. However, due to large variations between different observers (e.g., reported κ-coefficient was 0.3), the reliability of the fibrosis evaluation remains a challenge [79]. Fortunately, computer-aided diagnosis tools can minimize observer bias because of their higher consistency, reproducibility, and standardization, as well as their ability to realize continuous quantification of fibrosis degree [80].

Ginley et al. reported a CNN model that was developed based on 116 WSIs [81]. With this model, the analysis results were not only close to the pathologist-determined scores of IF and TA but also significantly associated with patient outcomes. Recently, with advanced algorithms, the accuracy of AI in recognition of the finer structure and subtle changes in kidney biopsies has been improved, and the prediction power of allograft function has also been strengthened [82,83]. Thus, apart from the scoring of fibrosis extent, AI technologies may play a more crucial role in the prognosis and monitoring of post-transplant patients.

#### 4.2.2. Lupus Nephritis

Lupus nephritis (LN) is characterized by the deposition of circulating or localized immune complexes in the kidneys [84]. Due to a deficiency in clinical manifestation of the pathological changes of LN, renal biopsies are recommended for all patients with LN to determine their pathological type [85]. The examination results of a renal biopsy are also closely associated with the formulation of immunotherapy regimens and a precise prognosis [86]. LN histology is routinely classified as Type I to VI based on the National Institutes of Health (NIH) Activity Index (NIH-AI) and NIH Chronicity Index (NIH-CI) for quantification of the degree of active inflammation and chronic changes as described in the International Society of Nephrology/Renal Pathology Society (ISN/RPS) 2018 classification [38]. However, some studies have pointed out that the classification criteria mentioned above were prone to interobserver variability with agreement ranging from poor to moderate [87]. Therefore, AI application tools can be adopted to improve the efficiency, objectivity, and accuracy of pathological diagnosis under the current guidelines.

A CNN model used to classify glomerular lesions in LN (impairments with slight/high severity or sclerosis) was developed by Zheng’s team using images obtained from 349 PAS-stained human kidney biopsies [88]. This model achieved a mean average precision of 0.807 at the glomerular level and attained a high concordance with the pathologist assessment at the kidney level (κ: 0.906). However, this model could only identify the most conspicuous lesions, which only account for a limited degree of LN pathological changes. Since some atypical LN lesions can be easily misclassified into other diseases, it is necessary to have a combinatorial assessment with other structural characteristics to distinguish the pathology.

Characteristic features of LN could also be identified using IF, including “full-house” staining and intensive C1q staining, as well as stained deposits outside the glomeruli or in the subendothelial and subepithelial layers [89]. Thus, some researchers tried to detect LN lesions in the IF background. For example, a multi-task learning (MTL) model was built to process IF images of four types of nephropathy [90]. The diagnosis of LN was improved by this model, with high accuracy of 0.91 and an AUC of 0.982, implying the promising potential for its clinical application in the future.

On the other hand, previous studies made attempts to integrate the baseline histopathological variables and laboratory data, which achieved remarkable advances in total accuracy and robustness [91,92]. Therefore, incorporating clinical indices into computer vision programs might overcome the limitations in the detection of LN lesions, thereby improving the accuracy of diagnosis and prognosis prediction.

#### 4.2.3. Diabetic Nephropathy

Diabetic nephropathy (DN), as the principal microvascular complication of diabetes, has become the primary factor leading to ESRD [93]. The main pathological changes of DN in renal biopsy samples are the diffuse mesangial expansion in the early stage and Kimmelstiel–Wilson nodule formation with longer diabetic duration. The most significant and earliest change observed using electron microscopy (EM) is glomerular basement membrane (GBM) thickening [94]. Thus, a pathological classification was proposed, which was used to describe the progression stages of DN according to the characteristic glomerular lesions [95]. However, the classification based on visual assessments by different pathologists may produce varied results. To address this issue, more efficient tools to assess disease severity are needed.

To improve diagnostic reproducibility, one research team tried to combine image analysis with CNN algorithms for the classification of renal biopsy samples collected from 54 DN patients [96]. The agreement between the AI model and ground truth in the quantification and classification of DN lesions achieved a moderate level (Cohen’s kappa: 0.55). Although the results showed that the AI-assisted tool is currently unable to replace the function of human pathologists, it is notable that beneficial attempts have been made to overcome the subjectivity of artificial classification and improve the accuracy of clinical decision-making workflows in DN diagnosis.

In addition to AI application in lesion recognition of PAS-stained images, ML methods could detect pathological changes in IF images to produce encouraging results. Although immune complex deposition is considered unrelated to the main pathogenesis of DN, a previous study confirmed the value of IF images in the pathological diagnosis of DN. In that study, IF images of 885 renal biopsies, stained for IgG, IgM, IgA, C1q, C3, and fibrinogen, were used to construct deep learning programs [97], the results of which revealed the better performance of the AI-assisted technique (AUC 1.00) compared with human eye observation (AUC 0.75833). Further visualization and interpretation demonstrated the advantage of AI for surveying the surrounding areas of DN glomeruli, especially with regards to its potential to identify new important sub-visual changes that could not be found with the human eye alone.

Recently, DN diagnosis using EM has contributed to breakthroughs made by AI-assisted technology. For example, Hacking et al. designed a deep learning model (the MedKidneyEM-v1 Classifier) to classify five different renal lesions, including diabetic glomerulosclerosis [98]. As expected, the performance of this model was excellent for identifying DN, with an accuracy of 88.89% and a recall rate of 66.67%. This pilot study not only confirmed the feasibility of the application of the deep learning model used for the analysis of EM images but also laid a technical foundation for the future development of AI-assisted EM models with optimized functions.

#### 4.2.4. IgA Nephropathy

IgA nephropathy (IgAN) is currently the most common primary glomerular disease in European and Asian populations, and approximately 30% of patients with IgAN ultimately progress to ESRD within 20–25 years [99,100]. Under a light microscope, IgAN may present with various pathological features, such as hypercellularity, the proliferation of the mesangial matrix, focal necrosis, and segmental glomerulosclerosis [101].

In 2016, the International IgA Nephropathy Network and the Renal Pathology Society issued revised Oxford classification criteria that included hypercellularity in the mesangium (M), endocapillary proliferation (E), segmental sclerosis and adhesion (S), tubular atrophy, and interstitial fibrosis (T), as well as cellular/fibrocellular crescent formation (C) [102]. Although the predictive value of the Oxford classification or MEST-C score for IgAN has been verified by many clinical studies [103,104,105], the cumbersome requirements of the Oxford classification still cost pathologists much time and energy. Moreover, classifying the pathological conditions of the glomerulus following the Oxford classification could be difficult for clinicians, although it is an important determinant of treatment strategy [47]. Therefore, the application of AI-assisted tools in the quantitative analysis and automatic scoring of IgAN images might help relieve the burdens of pathologists and improve the accuracy of diagnosis.

In 2020, Zeng et al. developed algorithms that were used to identify glomerular lesions based on renal biopsy images collected from over 400 IgAN patients [106]. Like previously established networks, the new AI-assisted models can carry out multi-tasks, such as automatic localization of the glomeruli and classification of basic glomerular lesions related to IgAN pathological changes, including glomerular sclerosis, segmental sclerosis, and crescents. Analysis performed by those models achieved about 93.1% precision and 92.8% accuracy. Notably, these models can also accurately identify resident cells, such as mesangial cells, endothelial cells, and podocytes in the glomeruli and also generate the corresponding M-score according to the ratio of these cells, implying the innovative function of AI application in the analytic renal pathology system (ARPS).

To further clarify the link between glomerular lesions and clinical indicators, Sato et al. proposed an unsupervised model integrated with CNN and a visualization algorithm, which can perform cluster analysis in renal biopsy specimens collected from 68 IgAN patients [107]. This model could classify the glomeruli into 12 types and 10 patches, upon analysis of which the corresponding histological score for each glomerulus or patient was calculated. This study confirmed the significant relationship between image-based scores and assessed clinical variables, although this new approach is not currently being applied in nephropathy. For instance, the defined sclerotic glomeruli were found to be associated with serum creatinine, systolic blood pressure, and urinary protein. These results not only provide visual interpretation for the previous findings according to the Oxford classification but also offer new insights into the relationship between pathological images and clinical variables [108].

Taken together, despite obvious progression in the diagnosis of IgAN with AI-assisted technology, there still exist several limitations in this field. The recognition scope is mainly restricted to glomerular lesions rather than the lesions involving the renal tubule and interstitium. Moreover, the methods for fine identification of histological structures, such as glomerular resident cells, are still in their infancy. Therefore, a deep investigation is required to improve the robustness and accuracy of these models. There are more requirements for realizing automatic Oxford scoring on all fronts. The summarization of auxiliary diagnosis via AI tools associated with specific kidney diseases is provided in Table 2.

**Table 2 jcm-11-04918-t002:** AI-assisted diagnosis of specific nephropathy.

Disease	Author	Year	Task	Methods	Slides	Main Results	Ref.
**Renal interstitial fibrosis**	Ginley et al.	2021	Detection and quantification of IFTA and glomerulosclerosis	CNN	116 WSIs, human (PAS)	High levels of agreement between CNN and four renal pathologists: IFTA agreement: ICC: 0.97 (0.94–0.99) glomerulosclerosis agreement: ICC: 0.91 (0.84–0.96)	[81]
	Marechal et al.	2022	Automated segmentation of kidney tissue	CNN	241 samples of healthy kidney tissue, human	AUC: tubular atrophy: 0.92 interstitial fibrosis level: 0.91 vascular luminal stenosis (>50%): 0.85	[82]
	Z. Yi et al.	2022	Recognition of interstitial fibrosis, tubular atrophy, and mononuclear leukocyte infiltration	U-Net and mask R-CNN algorithms	789 transplant biopsies, human (PAS)	Recognition of abnormal tubules: TPR: 84%	[83]
	Farris et al.	2021	Quantification of interstitial fibrosis	VGG19 CNN	100 biopsy specimens, human	Moderate agreement between algorithm and pathologists: correlation coefficient: 0.46 (0.40–0.52)	[109]
**Lupus nephritis**	Yang et al.	2021	Identification of glomerular lesion	ResNeXt-101	146 class III or IV (±class V) lupus nephritis biopsies, human (H&E)	Identification of globally sclerotic glomeruli: accuracy: 0.98–0.99 AUC of each kind of lesion: 0.687–0.946	[66]
	Zheng et al.	2021	Classification of glomerular pathological findings in LN	YOLOv4 and VGG16	349 annotated WSIs (PAS) 321 unannotated WSIs (PAS)	Glomerular level: F1 (“slight” and “severe”): 0.924–0.952 Per-patient kidney level: weighted kappa with nephropathologist: 0.855	[88]
	Pan et al.	2021	Classification of kidney diseases in IF images	AlexNet	655 IF images of IgAN (IF)	AUC of non-blurred IF images: 0.997 AUC of blurred IF images: 0.992	[90]
	Cicalese et al.	2021	Classification of LGN	Uncertainty-guided Bayesian classification scheme	87 biopsy specimens, mice (PAS)	Weighted glomerular-level accuracy: 94.5%, weighted kidney-level accuracy: 96.6%	[110]
**Diabetic nephropathy**	Ginley B et al.	2019	Classification of glomerular lesions	CNN	54 WSIs, human (PAS); 24 WSIs, mice (PAS)	Moderate Cohen’s kappa κ of agreement with a senior pathologist: 0.55 (0.40–0.60)	[96]
	Kitamura S et al.	2020	Diagnosis of diabetic nephropathy with renal pathological immunofluorescence	Deep learning	885 renal immunofluorescent images, human	Six programs showed 100% accuracy, precision, and recall, and the AUC was 1.000	[97]
	Hacking S et al.	2021	Classification of medical kidney disease on electron microscopy images	MedKidneyEM-v1 classifier (deep learning)	600 images	Diabetic glomerulosclerosis: precision: 88.89% recall: 66.67%	[98]
	Ravi et al.	2019	Detection of glomerulosclerosis in DN	Genetic k-means	-	Detect 99% of pathological DN glomerulosclerosis	[111]
**IgA nephropathy**	Zeng et al.	2020	Identification of glomerular lesions and intrinsic glomerular cell types	ARPS	400 WSIs, human (PAS)	Evaluation of global, segmental glomerular sclerosis, and crescents: Cohen’s kappa values: 1.0, 0.776, 0.861	[106]
	Sato N et al.	2021	Evaluation of the relationship between kidney histological images and clinical information	CNN	68 WSIs, human (H&E)	Significant relationship between the score of the patch-based cluster containing crescentic glomeruli and SCr: coefficient = 0.09, *p* = 0.019	[107]
	Purwar R et al.	2022	Detection of mesangial hypercellularity of MEST-C score	CNN	138 individual glomerulus images of IgA patients	Accuracy: 90 ± 2%, sensitivity: 90.4%, specificity: 80%	[112]

Abbreviations: CNN: convolutional neural network; PAS: periodic acid–Schiff; IFTA: interstitial fibrosis, tubular atrophy; ICC: intraclass correlation coefficient; AUC: area under the curve; TPR: true positive rates; LGN: lupus glomerulonephritis; DN: diabetic nephropathy; ARPS: analytic renal pathology system.

### 4.3. Prognosis Prediction

Aside from its application in structure identification and auxiliary pathological diagnosis, AI-assisted technology can be applied in other tasks, such as prognosis prediction based on pathological images, risk stratification, and evaluation of therapeutic outcomes.

As an early stage exploration, Lee’s team used the unsupervised learning method to predict baseline and 1-year changes in the estimated glomerular filtration rate (eGFR). Based on the comprehensive morphological features extracted from 161 renal biopsies, along with patient clinical information, the AUC of the prediction model for eGFR at biopsy time reached 0.93, while that for 1-year eGFR was 0.80. These results indicated the potential of visual-feature-based algorithms for predicting CKD progression [113].

With regards to the prediction for specific kidney diseases such as interstitial fibrosis, Kolachalama et al. trained a CNN model to predict the renal survival rates at 1, 3, and 5 years based on the images of trichrome-stained renal biopsies with varying degrees of fibrosis. The AI-aided prediction tools achieved higher AUCs than did human pathologists, suggesting the feasibility of the image-based AI models in clinical decision-making augmentation [41]. The examples of digital image-based prediction for renal prognosis using AI algorithms are described in Table 3.

**Table 3 jcm-11-04918-t003:** Auxiliary prediction for prognosis.

Author	Year	Task	Methods	Slides	Main results	Ref.
Kolachalama et al.	2018	Prediction of the 1-, 3-, and 5-year renal survival rates	CNN	300 biopsies, human (trichrome-stain)	AUC of 1-, 3-, and 5-year renal survival: 0.878, 0.875, and 0.904	[41]
Lee et al.	2022	Prediction of the baseline eGFR and 1-year change	ML	161 biopsies human (trichrome-stain)	AUC of baseline eGFR: 0.93, AUC of 1-year eGFR: 0.80	[113]
Ledbetter et al.	2017	Prediction of 1-year eGFR	CNN	80 biopsies, human (trichrome-stain, PAS)	Mean absolute error of 17.55 mL/min	[114]

Abbreviations: CNN: convolutional neural network; eGFR: estimated glomerular filtration rate; ML: machine learning.

## 5. Challenges and Limitations

Although the AI-assisted technology is superior to the human eye for identifying certain kidney structures in histological images, challenges still exist for its large-scale application in nephropathy due to the limitations described below.

### 5.1. Lack of Accountability

Lack of accountability is a common problem for machine learning algorithms. After training, artificial neural networks can automatically extract image features from large datasets for a specific purpose. However, the specific process of defining the image features by self-training remains a black box. Some studies reported the paradoxical phenomenon that imperceptible modification of images led to significant degradation of network performance, while major changes in the same images could not influence the classification results [115,116]. The process for ML algorithms to make judgments remains largely unknown, so the reliability of AI-assisted diagnosis is still being questioned.

### 5.2. Insufficient Data

Another limitation arises from the lack of standard datasets used for training, which is critical for the effectiveness of CNN algorithms. Small-scale or poor-quality datasets might impair the efficiency and accuracy of ML algorithms. However, in the field of renal pathology, few studies can obtain massive image samples for training purposes, because of the relative rarity of kidney diseases and the considerable cost (labor and time) for manual annotation by skilled nephropathologists.

### 5.3. Variations of the Image Quality

Unexpected variations in the input images constitute another obstacle for AI application in nephropathology [117]. The lack of a standardized processing workflow of WSIs may be caused by variable staining methods, staining time, scanning equipment, and file format, all of which could affect the judgment of the algorithms. Hence, to further expand the application scope of AI in nephropathology, a unified criteria for sample processing needs to be established, and an integrated information system with a consistent file format needs to be built [118].

## 6. Outlook for the Future

AI has profoundly changed the traditional clinical models used in the past decades, especially in oncology and radiology. Unfortunately, compared with flourishing advancements in other fields such as hepatology and neurology, AI-assisted technology in nephrology is still immature [119]. The tremendous diversity in disease progression, outcomes, and responses to therapy also increases the difficulty of algorithm design, which becomes an obstacle to the implementation of AI-assisted technology in nephropathology [120]. Despite the challenges, opportunities also await. The following sections discuss the possible future directions for the application of AI-assisted technology in nephropathology.

### 6.1. Fusion of Data

Present ML methods that are mainly based on a single type of staining could achieve high accuracy for identifying a single disease; however, they are unable to distinguish mixed diseases. To solve this problem, Vasiljević et al. proposed an unsupervised learning algorithm to achieve “image translation” between different staining methods, which has proven useful in overcoming the barriers of inter-staining fusion [121]. Multi-modal learning, which refers to the combination of different sources of information [122], including pathological results, images, clinical history, and biochemical test results, has gained early success. An automated workflow was recently developed for integrating multi-modal data of mouse models of pancreatic cancer, which can transfer annotations between histology data and MSI data [123]. Therefore, in the field of renal pathology, it may also be possible to apply this technology in combination with pathological images and other “omics” data, thereby increasing the accuracy of prognosis prediction.

### 6.2. Application of State-of-the-Art Technology

With the rapid development of AI-assisted technology, original algorithms could be optimized, and new learning theories are proposed to solve complex practical problems. For example, transfer learning is considered an effective tool to overcome the limitation of data scale through the strategy of pre-mining knowledge on a related dataset with large volumes of data [124]. In terms of renal pathology, a recent study successfully transferred the learning experience acquired from experimental animal models to the identification of human specimens [69]. Similarly, the multi-task learning algorithm also provides a solution to limited data. Through its stronger generalization ability to integrate relevant tasks for sharing obtained features, the multi-task learning requires less training data and significantly improves the overall performance [125]. A recently published report revealed that both segmentation of rectal cancer lesions and prediction for the response to neoadjuvant chemoradiotherapy can be implemented at the same time using a multi-task learning method [126]. Thus, this novel technology is expected to make a difference in the field of nephropathology.

### 6.3. Make Full Use of the Unknown

A lack of awareness of the exact mechanisms underlying the deep learning process presents a significant challenge. Nevertheless, AI-assisted technology can extract sub-visual features from images with billions of pixels, thereby successfully detecting subtle pathological changes that may be neglected by the human eye. For example, for a long time, IF images were considered to be nonspecific for making a diagnosis of diabetic nephropathy. However, a recent study reversed this opinion by developing an AI-assisted algorithm based on IF images, showing high accuracy and AUC of this method [97].

In addition, the development of semi-supervised and unsupervised learning algorithms also enables the computer to explore the natural patterns of the data without the interference of predefined information [127]. For instance, utilizing unsupervised clustering analysis, IgA glomerular lesions can be automatically classified, and a new histological scoring system can be established to integrate the clinical data and pathological image changes [108]. Therefore, AI-assisted technology can promote the discovery of new features and help latent associations to be revealed in the field of renal pathology.

### 6.4. Association of AI with Nephrologists

In the era of AI, coordination of the relationship between computers and human beings is becoming common. Compared with a substitute, AI is more like an auxiliary tool to enhance our intelligence [128]. The application of AI-assisted technology in nephropathology can help pathologists to accomplish repetitive tasks, such as counting the numbers of glomeruli and providing diagnosis references for nephrologists. However, at the current stage, the final diagnosis still requires verification by experienced pathologists. Furthermore, the development and refinement of AI algorithms cannot be implemented without the involvement of pathologists. Thus, the application of AI-assisted technology in nephropathology is unable to replace renal pathologists, who still play an essential role in practice.

Moreover, nephrologists should embrace the new trend in AI-assisted diagnosis with a predisposition to learn and adapt, as well as be prepared to face the challenges and caution signs of hidden problems. With persistent efforts, the combination of clinical medicine and AI will produce more encouraging results and advance the field of personalized precision medicine in the future.

## Figures and Tables

**Figure 1 jcm-11-04918-f001:**
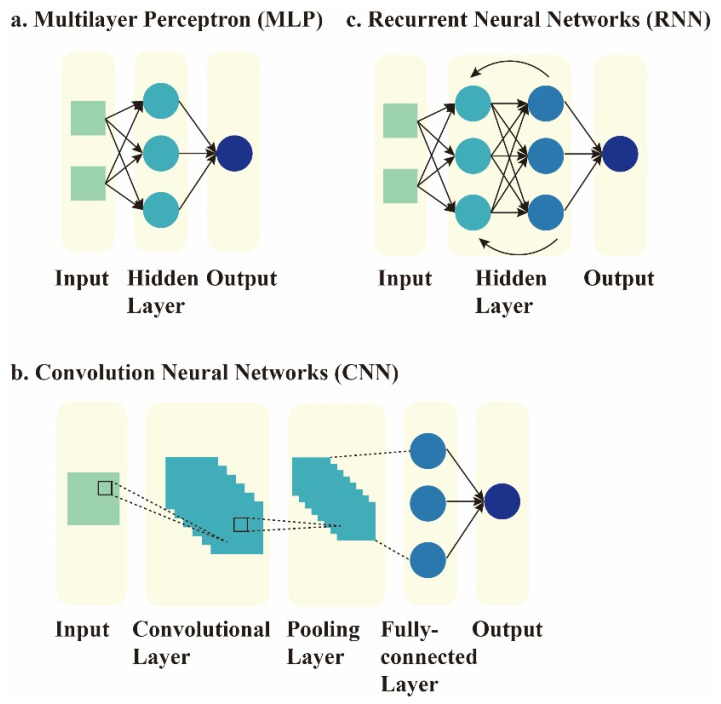
Visualization of three network architectures in deep learning. (**a**) The MLP is a basic feedforward artificial neural network, which consists of multiple fully connected layers, including the input layers, a stack of hidden layers, and output layers. Each new layer receives outputs weighted by the prior layer and directs the flow to the subsequent one. Backpropagation is applied for the iteration in order to obtain desired parameters. With less complicated architectures, MLP models require lower computing power, which is suitable for simple classification problems or nonlinear regression analysis [11]. (**b**) A CNN is composed of a series of layers with specific functions, such as convolution, nonlinear activation, and pooling. A CNN can reduce the high dimensionality of images. Analogous to the architecture of the visual cortex, each neuron in the convolution layer responds only to the filter-extracted area of the previous layer and overlaps with each other to cover the entire image. Thus, the convolutional layers enable the identification of important features with fewer parameters. Finally, the last few fully connected layers will process the condensed image information and obtain probabilities of the input belonging to a particular class. Employing relevant filters, parameter reduction, and weight reusability, CNNs can achieve more robust performance in analyzing complicated images with spatial and temporal dependencies [12]. Moreover, by its ability to learn features equivariantly, CNNs also have advantages in processing and differentiating similar images regardless of position and imaging condition variations [10]. (**c**) An RNN is characterized by cyclic connections that allow information to flow back and be preserved in its hidden layers. Thus, previous outputs can exert their influence on the current inputs and outputs. The RNN is applicable not only for sequentially related data (such as handwriting or speech-language) but also for information with an ordered spatial structure (such as image pixels). However, the RNN may not be suitable for long-time memory storage, because information by gradient will get lost rapidly over time [13].

**Figure 2 jcm-11-04918-f002:**
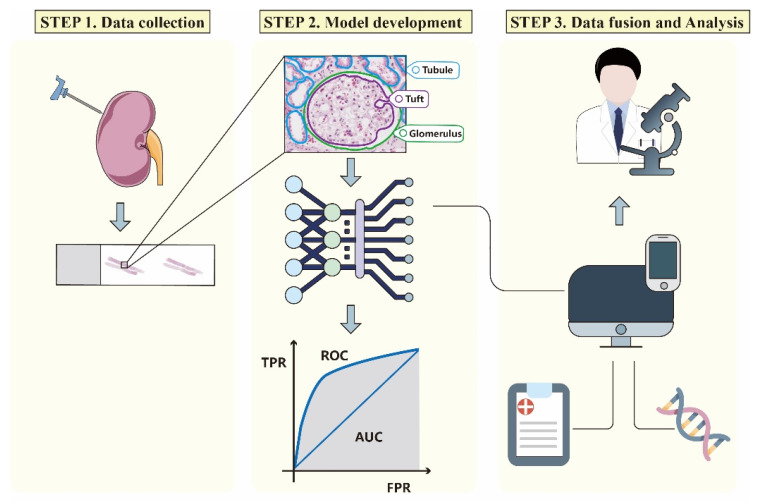
Simplified workflow for AI-assisted technology in renal pathology. First, kidney tissues are obtained by renal biopsies and scanned into WSIs for the subsequent analysis. Secondly, the digital images are divided into different parts manually with corresponding annotations and then transferred into the training model as inputs. The performance of the AI-assisted model is tested using another independent dataset to verify its robustness. Finally, different modalities of data including images, “omics”, and clinical information are integrated, which enables the model to make a more accurate judgment and provide valuable references for pathologists.

## Data Availability

Not applicable.

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
