# Peer review of "Artificial Intelligence-Assisted Renal Pathology: Advances and Prospects"

_jcm, 2022, doi:10.3390/jcm11164918_

Round 1

Reviewer 1 Report

The present review deals with artificial intelligence and renal pathology. The topic is nice and well written. Many medical condtions requiring renal biopsy may impact on renal pathology.

The role of artificial intelligence in distinguishing among the different histological sutypes of kidney cancer or simply between the benign or malignant aspect,  deserve a paragraph in the manuscript and some comments.

Author Response

Response to Reviewer 1

Comments and Suggestions for Authors

The present review deals with artificial intelligence and renal pathology. The topic is nice and well written. Many medical condtions requiring renal biopsy may impact on renal pathology.

Response: Thanks for your appreciation!

1.The role of artificial intelligence in distinguishing among the different histological subtypes of kidney cancer or simply between the benign or malignant aspect, deserve a paragraph in the manuscript and some comments.

Response: Thank you for pointing this out. We have added a paragraph to briefly illustrate the application of AI in kidney cancer detection.

The revised text reads as follows on page 6, lines 229-239:

“Moreover, as a special “structure” in kidney, cancer masses can also be automatically classified and evaluated by grades with the help of deep learning algorithms. Fenstermaker et al. developed a CNN model based on HE-stained WSIs from 42 renal cell carcinoma (RCC) specimens to distinguish normal tissue and 3 histology cancer subtypes including clear cell, chromophobe, and papillary carcinoma. The accuracy of the model could reach as high as 99% in tumor tissue identification and 97.5% in RCC subtype classification. In addition, the model also predicted prognosis-associated Fuhrman grade according to nuclear size and polymorphism with a high accuracy of 98.4%. Thus, the results indicate that by highlighting the region of interest in advance and presenting their judgements for reference, artificial intelligence methods are expected to improve the accuracy and efficiency of kidney cancer diagnosis in the future.”

Reviewer 2 Report

In this manuscript, Wang et al. explored the innovation of AI and related potential implementation in the diagnostic daily routine of kidney conditions. They provided a comprehensive overview of the AI fundamentals before introducing the major advances in this field.

Overall, the manuscript is well written, the approach is methodical and stepwise, and the topic is of interest and potential clinical impact.

However, there are some parts of the manuscript that are overstatement either of AI practical applications and pathology current limitations.

For example, pathologists do not struggle in “deciphering complicated digital images” (abstract, lines 13-14), but, contrarily, they usually provide helpful insights to tune AI algorithms and refine digital pathology development.

This Review (and AI in general) do not need sensationalist statements (i.e., “AI-based state-of-the-art technology could be applied to eradicate these deficiencies”, page 4, lines 156-157; or “there is an urgent need to introduce AI-assisted WSI analysis for precise evaluation of glomeruli”, page 5, lines 169-170), as the benefits are evident as well as the several limitations that are still present and prevent to date a full clinical implementation.

I strongly suggest the Authors to dampen the sensationalist “tone” of the manuscript (particularly evident in the first pages).

Also, provide larger images with more clear text, as the ones provided are difficult to read.

Author Response

Response to Reviewer 2

Comments of this reviewer are

In this manuscript, Wang et al. explored the innovation of AI and related potential implementation in the diagnostic daily routine of kidney conditions. They provided a comprehensive overview of the AI fundamentals before introducing the major advances in this field. Overall, the manuscript is well written, the approach is methodical and stepwise, and the topic is of interest and potential clinical impact.

Response: Thank you!

  1. However, there are some parts of the manuscript that are overstatement either of AI practical applications and pathology current limitations. For example, pathologists do not struggle in “deciphering complicated digital images” (abstract, lines 13-14), but, contrarily, they usually provide helpful insights to tune AI algorithms and refine digital pathology development.

Response: Thank you very much for your valuable suggestion. We have corrected some arbitrary statements in the text to provide a more objective and cautious opinion about artificial intelligence.

The “However, renal pathologists still face challenges in deciphering complicated digital images, especially when analyzing large amounts of digital pathology imaging data” has been deleted from the text (abstract, lines 13-14).

2.This Review (and AI in general) do not need sensationalist statements (i.e., “AI-based state-of-the-art technology could be applied to eradicate these deficiencies”, page 4, lines 156-157; or “there is an urgent need to introduce AI-assisted WSI analysis for precise evaluation of glomeruli”, page 5, lines 169-170), as the benefits are evident as well as the several limitations that are still present and prevent to date a full clinical implementation.

Response: Thank you for your kind suggestion. We have modified it.

The “AI-based state-of-the-art technology could be applied to eradicate these deficiencies” has been corrected as “AI-based state-of-the-art technology might provide a possible solution for this problem” (page 4, lines 155-156).

The “there is an urgent need to introduce AI-assisted WSI analysis for precise evaluation of glomeruli” has been corrected as “Therefore, the introduction of AI-assisted WSI analysis might be helpful to promote more precise evaluation of glomeruli.” (page 5, lines 168-170)

  1. I strongly suggest the Authors to dampen the sensationalist “tone” of the manuscript (particularly evident in the first pages).

Response: According to your suggestion, statements with sensationalist “tone” are also modified as follows:

“However, the current diagnosis made by renal pathology mainly depends on the assessment of a renal pathologist, which is not only time-consuming and labor-intensive but also involves subjectivity and relatively poor reproducibility” (page 4, lines 150-152).

“Therefore, AI application tools can be adopted to improve the efficiency, objectivity, and accuracy of pathological diagnosis under the current guidelines” (page 11, lines 288-289).

“Although the predictive value of the Oxford classification or MEST-C score for IgAN has been verified by many clinical studies, the cumbersome requirements of the Oxford classification still cost pathologists much time and energy pathologists.” (page 12, lines 359-362)

“Besides, the development and refinement of AI algorithms can’t be implemented without the involvement of pathologists. Thus, the application of AI-assisted technology in nephropathology is unable to replace renal pathologists, who still play an essential role in practice.” (page 17, lines 507-510)

  1. Also, provide larger images with more clear text, as the ones provided are difficult to read.

Response: Thank you for this suggestion. We have uploaded new images with larger size and bigger fonts.